# WhatsApp in Clinical Practice—The Challenges of Record Keeping and Storage. A Scoping Review

**DOI:** 10.3390/ijerph182413426

**Published:** 2021-12-20

**Authors:** Christopher Morris, Richard E. Scott, Maurice Mars

**Affiliations:** 1Department of TeleHealth, School of Nursing & Public Health, College of Health Sciences, University of KwaZulu-Natal, Durban 4041, South Africa; ntc.ehealthconsulting@gmail.com (R.E.S.); mars@ukzn.ac.za (M.M.); 2Department of Community Health Sciences, Cumming School of Medicine, University of Calgary, Calgary, AB T2N 4N1, Canada; 3Flinders Digital Health Research Centre, College of Nursing & Health Sciences, Flinders University, 5042 Tonsley, Australia

**Keywords:** WhatsApp, record keeping, medical records, data storage

## Abstract

The use of WhatsApp in health care has increased, especially since the COVID-19 pandemic, but there is a need to safeguard electronic patient information when incorporating it into a medical record, be it electronic or paper based. The aim of this study was to review the literature on how clinicians who use WhatsApp in clinical practice keep medical records of the content of WhatsApp messages and how they store WhatsApp messages and/or attachments. A scoping review of nine databases sought evidence of record keeping or data storage related to use of WhatsApp in clinical practice up to 31 December 2020. Sixteen of 346 papers met study criteria. Most clinicians were aware that they must comply with statutory reporting requirements in keeping medical records of all electronic communications. However, this study showed a general lack of awareness or concern about flaunting existing privacy and security legislation. No clear mechanisms for record keeping or data storage of WhatsApp content were provided. In the absence of clear guidelines, problematic practices and workarounds have been created, increasing legal, regulatory and ethical concerns. There is a need to raise awareness of the problems clinicians face in meeting these obligations and to urgently provide viable guidance.

## 1. Introduction

The use of instant messaging applications, and in particular WhatsApp, to share patient information between clinicians is becoming increasingly common [1,2,3]. An earlier review noted that most WhatsApp use in clinical services was in the developing world [4]. For example, studies from Malaysia, South Africa, and Brazil show that WhatsApp use is common (with 74%, 87%, and 97% of clinicians, respectively, using WhatsApp [5,6,7]), including for second opinions or sharing of patient information. Its use in the developed world is now also common [1,2,8], and has grown further during the COVID-19 pandemic, with searches of PubMed on WhatsApp showing a marked increase in papers: 2018—94; 2019—126; 2020—312; 2021 (to 30 August 2021)—323.

Record keeping and storage of medical records are a legal requirement in many countries [9], and as early as 1999, the World Medical Association (WMA) had made clinicians aware of the need to maintain clinical records of telemedicine consultations [10], reiterated in 2018 together with legal and ethical obligations to protect sensitive patient data [11]. The absence of clear guidelines when using social media apps such as WhatsApp [9] has created problematic practices and workarounds particularly for issues of record keeping and data storage, and only serves to increase legal, regulatory and ethical concerns for patient privacy and the safeguarding of protected health information.

In the absence of formal and broadly accepted definitions of record keeping and storage, functional definitions were developed and adopted, as follows. Record keeping is defined as: maintaining, for each patient, a contemporaneous, chronological, secure, attributable, legible, traceable, permanent, original, accurate and date and time-noted health care record, whether paper or electronic, that documents in sufficient detail all health care interactions. Storage is defined as: the safe retention of health care records, with enduring access for a defined retention period, filed in a suitable systematic and permanent form, such as (for paper records) books, binders, files, cards or folders, or (for electronic records) in digital form in accordance with pertinent local or national policies and guidelines with respect to the creation, maintenance, security, disposition and recovery of electronic records.

The growing use of WhatsApp is despite its use not being fully General Data Protection Regulation (GDPR) [12] or Health Insurance Portability and Accountability Act (HIPAA) [13] compliant. Patient information sent as text messages, photographs, or video may contain sensitive, private, health-related information, in the form of electronic data. Over 125 countries now have strict data protection laws or regulations [14], many of which impact health and health care. Examples include the GDPR in the European Union [15], Lei Geral de Proteção de Dados in Brazil [16], HIPAA in the United States [17], and the Protection of Personal Information Act (POPI) in South Africa [18]. HIPAA introduced the term electronic protected health information (ePHI) as any information in an electronic format that can be used to identify a patient, defined as “personal data related to the physical or mental health of a natural person, including the provision of health care services, which reveal information about his or her health status.”

Beyond data protection legislation, clinicians must also comply with jurisdictionally specific legal and ethical responsibilities to keep patient records and store patient information. Traditionally, clinicians have kept paper-based files and notes in the patient’s medical record stored in a filing cabinet and, more recently, in electronic records of one form or another (e.g., electronic health record (EHR), electronic medical record (EMR), and electronic patient record (EPR)).

With the growing use of smartphones and instant messaging apps such as WhatsApp, neither designed for medical use, patient information often resides on both the senders’ and recipients’ mobile phones. Proactive steps have to be taken to transfer the data to a medical record and to then store the electronic data. However, some consider the information to be “stored” on mobile phones and that this constitutes a “record” of the communication/s, thereby addressing both record keeping and storage [4]. The legal, regulatory and ethical risks of this approach and a framework for their mitigation has been proposed in the Cellphone Stewardship Framework for Health Care Providers (CSF-HCP) [19].

Privacy, data security, IT governance, and mobile phone stewardship issues, and the legal obligation to keep medical records, require that patient information be added to, or incorporated into, a medical record, be it electronic or paper based. At issue is, can the information be transferred to a medical record, how can this be done, and if having done so, is there a need to retain the original text message/s and attachments?

Aligning the use of WhatsApp with record keeping and storage of medical records is complex (complying with individual country/region legislation), and fluid (changing with adjustments to both legislation, and WhatsApp versions and privacy policy). The aim of this study was to review the literature on how clinicians who use WhatsApp in clinical practice keep medical records of the content of WhatsApp messages and how they store WhatsApp messages and/or attachments. The goal of this study is to inform and raise awareness of these and related issues and to encourage debate and resolution.

## 2. Methods

A scoping review was undertaken in accordance with published guidance [20]. Nine databases were searched, up to 31 December 2020, for articles on WhatsApp use in clinical practice: PubMed, Scopus, Science Direct and six databases within EbscoHost—CINAHL with full text, Health Source Nursing/academic edition, Index to legal periodicals, PsycARTICLES, PsycINFO and MEDLINE.

The search terms used varied according to database (Table 1). After duplicates were removed, titles and abstracts of the remaining resources were reviewed by all authors against inclusion and exclusion criteria, with resolution of any disagreements by consensus. Inclusion criteria were that the paper was in English, reported on WhatsApp in clinical use, and addressed record keeping or storage of WhatsApp messages and attachments. Book chapters, conference proceedings that were not full-length papers, and papers on the use of WhatsApp for behaviour change, education, appointment reminders or medication adherence were excluded. Full-text papers of the resources meeting the criteria were obtained and reviewed by all authors against the inclusion and exclusion criteria until final selection, with consensus. The information was charted in an Excel spreadsheet, and included record keeping and storage steps, country in which the study took place, and the medical discipline involved, and were then categorised by all authors.

## 3. Results

The searches yielded 2079 initial resources (Figure 1). After full-text review, 16 papers met the study criteria [5,21,22,23,24,25,26,27,28,29,30,31,32,33,34,35]. Of these, 11 reported on both record keeping and storage [5,21,22,23,24,25,26,27,28,29,30], four papers reported only on record keeping [31,32,33,34], and one paper reported only on storage [35]. Papers originated from Africa (4) [21,23,30,35], India (3) [22,24,25], EU (3) [28,33,34], UK (3) [26,27,32], Middle East (2) [29,31], and Asia (1) [5].

Ten papers reported the use of WhatsApp in surgical disciplines; general surgery [27,28,33,34], maxillofacial surgery [25,26], neurosurgery [22], otolaryngology surgery [29], plastic surgery [31], and orthopaedic surgery [32]. The remaining six papers were from a range of disciplines: dermatology [23,30], burns [21,35], paediatric patient transfer [24], and general medical and emergency services [5]. A summary and characteristics of included studies are contained in Appendix A.

The methods of record keeping and/or storage were grouped into five categories.

### 3.1. Group A. Prescribed Action—Electronic

Two papers reported transfer of data from mobile phones to electronic versions of patient notes or departmental records. Transfer was performed manually to a password protected database [21], or to a departmental secure computer [22], also presumably manually. One paper formally reported deleting messages off mobile phones after transfer [21], but this was only inferred from the descriptions in the other paper [22].

### 3.2. Group B. Prescribed Action—Paper Based

Three papers reported ‘downloading’ of a hard copy/script for record keeping before deleting data from ‘participant devices’ after a defined period of time [27,28,29], but the method was not described.

### 3.3. Group C. Prescribed Action—Uncertain Electronic or Paper Based

Four papers reported keeping records, but it was not clear from descriptions if this was done electronically or was paper based [24,31,32,35]. Three of these papers formally reported deleting messages off mobile phones [24,32,35]. In a burn service, all communications were removed once the clinical scenario had been addressed, and the importance of record keeping and storage was noted, but no details were provided on how this was done [35]. Ellanti et al. reported that data were deleted from each participant’s mobile phone after a 6 month period and although no mention was made of formal storage or record keeping, this was inferred from the descriptions in the paper [32].

Neogi and Panda reported keeping records of all patients physically (either analogue or digital) at the ‘referred hospital’ and periodically deleting all ‘archived data’ [24]. Another paper reported photographing a screenshot for saving in the medical record, but it was not clear if and how the screenshots were stored, but it seems unlikely they were printed as it was reported that WhatsApp conversations could not be printed [31].

### 3.4. Group D. Inaction—ePHI Remains on Mobile Phones

Some felt that information stored on the users’ mobile phone constituted a medical record [5,25,33,34]. Benefits of this were: a record of communication for audit and training purposes [5,27] and a digital record for future reference such that “lost X-rays are a thing of the past” [25]. In a dermatology service, some messages were stored on the specialists’ mobile phones [23].

### 3.5. Group E. Uncertain

Two papers mentioned but did not report evidence of record keeping or storage of WhatsApp message content [26,30]. Dungarwalla et al. acknowledged that records constituted a pillar of good clinical practice and governance but reported difficulties with transferring consults to patient records when using the departmental mobile phone [26]. Williams and Kovarik reported the inability to save data centrally or integrate WhatsApp consultations into a patient’s medical record [30]. Of note was that in four retrospective studies, information was accessed from WhatsApp messages stored on the users’ mobile phones [22,26,33,34] with no mention of subsequent deletion of messages.

## 4. Discussion

While the use of WhatsApp is becoming increasingly common [6,7], there are few papers reporting record keeping and storage of sensitive patient health information contained in WhatsApp messages and attachments. Only 16 papers mentioned keeping records and/or storing of data transmitted using WhatsApp. Further, there is no clear evidence from the reviewed literature as to how, when using WhatsApp, patient information can routinely be transferred to, or incorporated into, a print or electronic medical record to permit record keeping and storage. Surprisingly, there were no reports of copies of WhatsApp messages being sent by email for record keeping or subsequent entry into an electronic medical record, a feature available within WhatsApp.

The absence of clear guidelines on record keeping and data storage has, as previously noted [9], created problematic practices and workarounds that only serve to increase legal, regulatory and ethical concern for patient privacy and the safeguarding of protected health information. For example, there was a general sense of concern about sensitive patient data being shared and stored on mobile phones, but those papers reporting deletion of messages on users’ phones did not report the message being deleted from the sender’s phone after completion of the case or other specified time period. Patient privacy is at risk when sensitive data are stored on mobile phones, and such practice is common. The problem is not confined to the use of instant messaging but also pertains to clinical photographs. Of 300 French plastic surgeons, 50% stored clinical photographs on their smartphones, whilst in Australia, 46% of dermatologists surveyed stored images on smartphones with limited security measures [36]. In a Canadian survey of plastic surgery residents and physicians, 57% stored such images on their mobile phones, with 73% of these storing clinical images among their personal photos [36].

Furthermore, a mobile phone may be lost or stolen, or content may be inadvertently shared. A survey of plastic surgeons reported 26% of respondents had accidentally revealed a clinical image to family or friends [37]. A safeguard to minimise this type of risk such as password protection was reported in one paper [22]. The term mobile phone “stewardship” has been defined and is applied to the appropriate care and use of mobile phones by health care workers. Good mobile phone stewardship practice recommends that messages are deleted off both the sender’s and receiver’s mobile phones [19].

Some authors were more mindful of concerns of breaching patient confidentiality but were less forthcoming in declaring their storage practices and even used the data stored on mobile phones for retrospective studies of WhatsApp use [22,26,33,34]. In addition, clinicians used WhatsApp despite recognising non-compliance with privacy laws [1] and/or contravention of organisational policies [38]. The reasons proposed were a lack of training in compliance with regulations [2] and the need for guidelines [39]. There is a general lack of awareness or concern about flaunting existing privacy and security legislation, regulations or guidelines [1,2,36] because the benefits to the patient and physicians outweigh the difficulty of compliance.

Ideally, every institution and medical practice should have an IT Governance Policy or Rules and standard operating procedures for the use of instant messaging, which would include record keeping and data storage. The reality is that in the developed world, the literature indicates that they are being ignored; and in the developing world, few medical practices and institutions have IT Governance Policies. No paper reported WhatsApp use in compliance with an IT Governance Policy. Johnston et al. reported special dispensation was given by the hospital’s information compliance department for the use of WhatsApp, provided that patient identifiable data were not shared, hardcopy records of the messages were kept, and WhatsApp messages were deleted from the phones at the end of each week [27].

Different approaches to record keeping and storage are possible (Figure 2). The figure shows the basic options for transferring WhatsApp chats (and/or attachments) to print or electronic formats capable of long-term storage, each of which was reported in the identified literature. These options provide potential solutions.

The problematic practices and workarounds noted earlier relate to safeguarding of protected health information, in particular retention of original messaging, long-term storage, encryption, extra-jurisdictional record keeping and storage, consent for subsequent use, and anonymisation.

### 4.1. Retention of Original Messaging

Few reports noted any concern or need for retention of original text messages or attachments, e.g., for audit purposes, although some did consider retention on their mobile devices as long-term ‘storage’ for clinical purposes [5,25,33,34]. Certainly, the literature implies, and a scan of the web shows, storage options exist through ‘back-up’ and cloud storage for WhatsApp chats and attachments. However, these will be fraught with their own security and confidentiality issues, and their longevity is uncertain. Many countries require electronic medical records be kept for several years after the death of a patient, but just how long would a commercial entity such as WhatsApp be able, or willing, to guarantee retention?

### 4.2. Long-Term Storage

As workarounds, there are a number of options to print out WhatsApp chats or convert them to pdf documents, which would allow both ‘print’ and ‘electronic’ (email; upload) transfer to medical records, offering storage options. However, any transmission of a pdf file (e.g., via email or over a network) would also require compliance with security and confidentiality requirements.

### 4.3. Encryption

Since 31 March 2016, messages between WhatsApp users have been protected with an end-to-end encryption protocol so that third parties, including WhatsApp and Facebook, cannot read them; the messages can only be decrypted by the recipient’s mobile phone [40]. All types of WhatsApp messages (chats, group chats, images, videos, voice messages, files) and WhatsApp calls, and any associated sensitive patient information, are protected by this end-to-end encryption, yet use of WhatsApp remains non-compliant with GDPR and HIPAA [12,13]. Furthermore, content may still be vulnerable if used for other purposes *before being* encrypted or *after* being decrypted using WhatsApp.

Due to the constant upgrading of security measures by WhatsApp, a number of concerns reported in earlier papers regarding storage of WhatsApp messages (containing sensitive patient information) may be misleading [26,41,42,43,44,45,46].

### 4.4. Extra-Jurisdictional Record Keeping and Storage

Increasingly, countries are introducing laws about extra-jurisdictional storage of health data. For example, the GDPR does not allow the storage of sensitive data of EU citizens on servers located outside the geographic area of the European Community [46]. Thus, WhatsApp messages are transmitted (and potentially stored for up to 30 days awaiting delivery) via servers located in the US, which may not comply with a particular country’s data protection regulations [26,45]. There has been concern about WhatsApp accessing and sharing information on users’ phones, however, this concern may be moot. The information gathered by WhatsApp and stored in the US on their servers is not ePHI, but contact information and possibly images if backed up to the cloud by the user. By downloading and using WhatsApp all users have knowingly or unwittingly consented to allow Facebook to access and download the telephonic contact details stored on their mobile phone. WhatsApp does share contact details with their parent company, Facebook, but it is important to emphasise that WhatsApp only stores users’ contact details, for which consent has been given when first downloading the app. When clinicians are sending messages to each other, WhatsApp is not able to access the patient’s contact details. Should a patient and clinician communicate directly with one another the patient’s contact details will already have been accessed by WhatsApp.

WhatsApp’s current and updated privacy policy allows Facebook to process additional user data that it collects from WhatsApp and importantly does not permit users, except within the EU, to opt out of accepting this policy. This “take it or leave it” privacy policy has caused concern in a number of countries, who are trying to negotiate an exemption from the policy [47].

### 4.5. Consent

Legislation in many countries require patient information be used only for the purposes for which consent was originally given, a common ethical principle. Thus, it may be a legal or ethical requirement that a patient give specific written informed consent before sensitive patient information is shared with another health professional or chat group of health professionals. Only one of the reviewed papers mentioned the need for keeping a record of informed consent (for example submitting a photograph of the signed consent) [21], although other sources acknowledged the need for consent [24,31] even if only verbal [22,35].

A recent review of consent practices when using WhatsApp found only 18 papers that reported obtaining either written or verbal consent for sharing information and/or images [48]. At one academic hospital, 97% of doctors surveyed did not obtain consent for sharing patient information by instant messaging [49]. Medico-legal providers recommend documenting consent in the patient’s notes when sharing images on mobile phones [1].

### 4.6. Anonymisation

Some have considered the use of WhatsApp to be permissible if sensitive patient details were not disclosed [26], and possible workarounds to comply with privacy and data security requirements were also suggested, primarily de-identification of patient information [50] and anonymisation [32]. However, this leads to an untenable conflict between ‘de-identification’ of messages and transfer to some form of ‘record’ and highlights the futility of such attempts. There is a spectrum of how anonymised personal health data may be, for example, use of medical record numbers or bed numbers [32], but truly anonymous (or anonymised) data are unacceptable in a clinical setting, where repeated confirmation of identity is the norm. Consider a clinician receiving anonymised data; how could any identifiable record be created from such anonymised data? Retaining the integrity of patient identity is crucial to safe health care delivery, and anonymisation is the antithesis. Merging of electronic (and paper) health records can and does occur but only under strict guidelines that require commonality of key identifiers. Once de-identified, merging is forever precluded.

The need for a WhatsApp-like instant messaging app for the health care sector has been identified [45]. Other instant messaging applications that meet HIPAA and/or GDPR requirements are available: Siilo, Hospify, Simple Practice, Oncall Health, Tiger connect, Trillian and MedX (for Australian registered doctors) [1,46]. In the UK, although Hospify is approved by the NHS, the use of WhatsApp and Telegram has recently been sanctioned “where there is no practical alternative and the benefits outweigh the risks” [51]. Each has strengths and weaknesses.

Study limitations are that while nine databases were searched, the grey literature was not searched (e.g., Google Scholar). Additionally, searches were restricted to the English language.

## 5. Summary

Only 16 of 346 papers reporting the use of WhatsApp in clinical practice addressed either record keeping or data storage. Most clinicians were aware that they must comply with statutory reporting requirements in keeping medical records of all electronic communications. For example, it was reported that records “constitute a pillar of good clinical practice and governance” [26] and that there was a need “for proper documentation in the medical record of valuable data and the content of consultations and treatment plans” [29]. Yet, it is clear that clinicians are failing to meet many legal, ethical, and good practice requirements. The reasons seem clear: on the one hand, WhatsApp is ubiquitous, freely available, easy to use, convenient, and meets clinicians’ needs. On the other hand, there is no comprehensive, consistent, and comprehensible guidance found in the literature [9] on the acceptable use of WhatsApp, nor how to transfer WhatsApp communications to a print or electronic patient record to allow satisfactory record keeping and storage [9].

There also remain untested limits to existing legislation. For example, the GDPR contains sections limiting the application of other restrictive sections when communications are for “preventive or occupational medicine, … medical diagnosis, the provision of health or social care or treatment, …”, and when the data “are processed by or under the responsibility of a professional subject to the obligation of professional secrecy under Union or Member State law or rules established by national competent bodies …” [52]. Could current concern be greater than required?

WhatsApp is also regularly upgraded and a number of concerns about data security related to message encryption, data transmission and data storage on external servers reported in earlier papers have been resolved. Of current concern is information contained in users’ contact lists, names, addresses and phone numbers being collected and used by Facebook. By downloading and using WhatsApp all users have knowingly or unwittingly consented to this, but the people whose information is being shared do not know with whom their information will be shared and how it will be used. This contravenes most existing data protection laws and regulations. The situation remains fluid and in the EU users can opt out of data sharing. A shortcoming of reported literature is the lack of clear statement about which WhatsApp application is being used. For example, WhatsApp for Business should not be used for health care as decrypted messages can be stored on external servers [12]. WhatsApp meets HIPAA requirements for data security during transmission; however, if information is stored on the phone, it becomes non-compliant as the app is not password protected, and the audit trail cannot be ensured as the user can delete the message [13].

Overall, the literature is confusing due to misinterpretation, misinformation, and constant updates to software versions and security protocols, and the introduction of new legislation. Users need to be made aware of the potential implications of the options they choose for record keeping and data and image storage, which may not be appropriate from a legal, regulatory, or ethical standpoint. Combinations and permutations of transmission for record keeping and storage are many. In general, unless specific choices have been made within WhatsApp or a user’s mobile phone to upload or back-up text messages, use of WhatsApp for general communication is secure. However, currently, there are no simple ‘GDPR/HIPAA proof’ solutions to record keeping or storage of WhatsApp content.

## 6. Conclusions

The findings of this study are telling. Despite the widespread use of WhatsApp, clinicians are either failing in their legal, regulatory, ethical, and clinical responsibility to keep records of WhatsApp consults, or are not reporting that they do so, nor how they do so. The literature does not report any clear “best practices” for record keeping or the secure storage of patient information obtained using WhatsApp. There is a need to raise awareness of the problems clinicians face in meeting these obligations and to urgently provide viable guidance.

## Figures and Tables

**Figure 1 ijerph-18-13426-f001:**
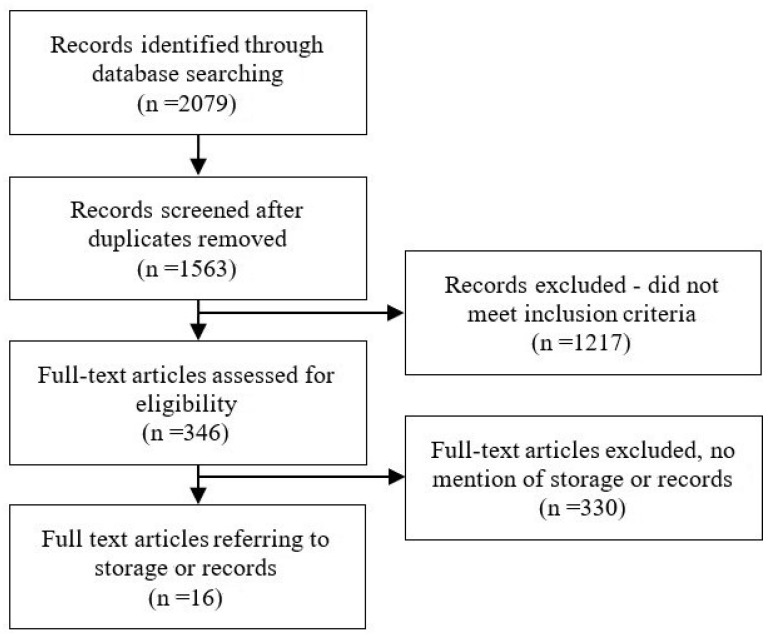
PRISMA Flow Diagram of the Search Process.

**Figure 2 ijerph-18-13426-f002:**
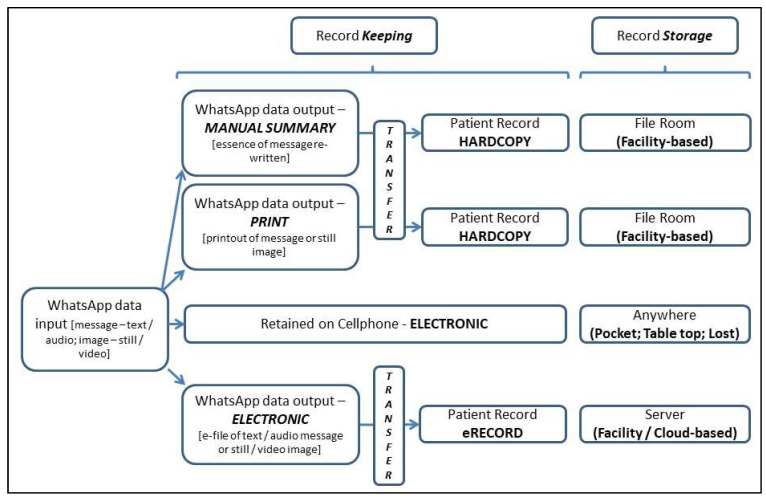
Options for Record Keeping and Storage of WhatsApp Messages.

**Table 1 ijerph-18-13426-t001:** Database, Search Strings and Resources Used for the Searches.

Database	Search Strings	Resources
PubMed	“WhatsApp” [All fields] ^a^	601
Scopus	(ALL (“WhatsApp”) AND ALL (“telemedicine” OR “telehealth” OR “ehealth” OR “e-health” OR “mhealth” OR “m-health”))	741
Science Direct	((“WhatsApp”) AND (“telemedicine” OR “telehealth” OR “ehealth” OR “e-health” OR “mhealth” OR “m-health”)) All fields	282
Ebsco Host	((“WhatsApp”) AND (“telemedicine” OR “telehealth” OR “ehealth” OR “e-health” OR “mhealth” OR “m-health”)) All text	503

^a^ No search modifiers used; PubMed is a biomedical and life sciences specific database.

## Data Availability

Not applicable.

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
