# Peer review of "WhatsApp in Clinical Practice—The Challenges of Record Keeping and Storage. A Scoping Review"

_ijerph, 2021, doi:10.3390/ijerph182413426_

Round 1
Reviewer 1 Report
Thank you for the opportunity to review this article which presents a scoping review of record keeping and storage of WhatsApp content in clinical practice. The article represents thorough work and is well written. I have some suggestions for improvements.
In the introduction, I would recommend a definition of the two key terms "record keeping" and "storage".
In the results, I would recommend including a table with an overview of included studies, with information about year of publication, first author, country / region, record keeping and storage. Furthermore, it may be useful for the reader to know what the use of WhatsApp was in the studies. Was it structured and planned use as part of the patient follow-up, or was it unstructured and random use. I think such information will help to get a comprehensive picture of practices related to WhatsApp use. In Figure 1, I would like to see the number of studies in each of the boxes.
I think the discussion is long, and recommend moving some of the text to the introduction.
Reviewer 2 Report
The challenge faced by undertaking a scoping type of review such as this is that the time lag for the publication of manuscripts in the scientific literature means that the information that you have gleaned by undertaking this scoping review is subject to limitations (which are not adequately discussed in the literature)
Of course what is not truly known at this point in time is how widespread the use of social media messaging apps truly is. So the information you have obtained may/may not reflect the current situation depending on the context. It would appear currently that these types of messaging apps are mainly used for conveying messages between healthcare workers that do not contain patient information within them (which is not the impression one gets from reading the Introduction section of the manuscript)
Currently there are governance frameworks in many (but not all) countries for the use of social media by Healthcare professionals. WhatsApp is a social media type messaging application and is in fact not the only one in existence. More needs to be made of what governance frameworks exist and hence what are the responsibilities of healthcare organisations to ensure that their employees comply. Much of what you mention in the results reflects governance failures and/or lack of knowledge, each of which is deserving of far more discussion for eg https://www.hipaajournal.com/hipaa-social-media/
Of note failure to comply with social media directives has proven costly to date for healthcare workers if they are caught transgressing them, which perhaps should also be mentioned
Plus the results may also lend themselves to being summarized into a Table for ease of reading
Finally you have not mentioned the other solution which is now being developed within healthcare systems (in light of the concerns about WhatsApp) and that is the development and implementation of messaging apps within healthcare systems which are regulatory compliant (for eg HIPAA compliant).
Round 2
Reviewer 2 Report
Hence there are significant limitations with the results of this scoping review which I believe are not adequately addressed in the discussion section of the manuscript. Of note I do have real world experience with the use of Whats App during recent times and there is no doubt in my mind that in regions where there are the requisite governance processes as well as comprehensive operational protocols then much of what is described in this particular manuscript about the problems with the use of WhatsApp can be mitigated. Hence what is described in the literature to date tends to reflect a variation in practice that tends to occur in settings where there is inadequate governance, limited regulatory oversight along with insufficient attention being paid to educating health care workers on its use.
Plus the difficulty with trying to review practices outlined in a number of manuscripts previously published from around the world such as has been undertaken in this particular instance is that there are significant constraints for the conclusions that one can draw due to the lack of standardized reporting (which is evident when you look at the tabulated information in Appendix 1).
Round 3
Reviewer 2 Report
I am happy with the changes which have been made. I have no other concerns